# No Correlation between PD-L1 and NIS Expression in Lymph Node Metastatic Papillary Thyroid Carcinoma

**DOI:** 10.3390/diagnostics14171858

**Published:** 2024-08-26

**Authors:** Lévay Bernadett, Kiss Alexandra, Fröhlich Georgina, Tóth Erika, Slezák András, Péter Ilona, Oberna Ferenc, Dohán Orsolya

**Affiliations:** 1National Institute of Oncology, Multidisciplinary Head and Neck Cancer Center, 1124 Budapest, Hungary; kissa@oncol.hu (K.A.); oberna.ferenc@oncol.hu (O.F.); 2National Institute of Oncology, Budapest, Center of Radiotherapy, 1124 Budapest, Hungary; turul49@yahoo.com; 3National Institute of Oncology, Department of Molecular Pathology and Surgical Pathology Center, 1124 Budapest, Hungary; dr.toth.erika@oncol.hu (T.E.); slezak.andras@oncol.hu (S.A.); helena@oncol.hu (P.I.); 4Department of Internal Medicine and Clinical Oncology, Semmelweis University, 1124 Budapest, Hungary; odohan@gmail.com

**Keywords:** thyroid papillary cancer, PD-L1 expression, NIS expression

## Abstract

Approximately 90% of thyroid cancers are differentiated thyroid cancers (DTCs), originating from follicular epithelial cells. Out of these, 90% are papillary thyroid cancer (PTC), and 10% are follicular thyroid cancer (FTC). The standard care procedure for PTC includes surgery, followed by radioiodine (RAI) ablation and thyroid-stimulating hormone (TSH) suppressive therapy. Globally, treating radioiodine-refractory DTC poses a challenge. During malignant transformation, thyroid epithelial cells often lose their ability to absorb radioiodine due to impaired membrane targeting or lack of NIS (sodium/iodide symporter) expression. Recent reports show an increase in PD-L1 (programmed death ligand 1) expression in thyroid cancer cells during dedifferentiation. However, no research exists wherein NIS and PD-L1 expression are analyzed together in thyroid cancer. Therefore, we aimed to investigate and correlate PD-L1 and NIS expression within primary tumor samples of lymph node metastatic PTC. We analyzed the expression of hNIS (human sodium/iodide symporter) and PD-L1 in primary tumor samples from metastatic PTC patients using immunohistochemistry. Immunohistochemistry analysis of PD-L1 and NIS was conducted in 89 and 86 PTC cases, respectively. Any subcellular NIS localization was counted as a positive result. PD-L1 expression was absent in 25 tumors, while 58 tumors displayed PD-L1 expression in 1–50% of their cells; in 6 tumors, over 50% of the cells tested positive for PD-L1. NIS immunohistochemistry was performed for 86 primary papillary carcinomas, with 51 out of 86 tumors showcasing NIS expression. Only in seven cases was NIS localized in the plasma membrane; in most tumors, NIS was primarily found in the intracytoplasmic membrane compartments. In the case of PD-L1 staining, cells showing linear membrane positivity of any intensity were counted as positive. The evaluation of NIS immunostaining was simpler: cells showing staining of any intensity of cytoplasmic or membranous fashion were counted as positive. The number of NIS positive cells can be further divided into cytoplasmic and membrane positive compartments. There was no observed correlation between PD-L1 and NIS expression. We can speculate that the manipulation of the PD-1/PD-L1 axis using anti-PD-L1 or anti-PD-1 antibodies could reinstate the functional expression of NIS. However, based on our study, the only conclusion that can be drawn is that there is no correlation between the percentage of NIS- or PD-L1-expressing tumor cells in the primary tumor of lymph node metastatic PTC.

## 1. Introduction

Thyroid cancer is the most common endocrine disease and represents 1% to 4% of all malignancies. The clinical behavior of thyroid cancer is highly variable, from indolent, slowly progressing tumors to aggressive ones with high mortality rates. Well-known that the thyroid gland is the most frequently involved in autoimmunity and cancer [1]. Clinical practice guidelines around the management of thyroid cancer have been modified over the last 15 years, with the goal of reducing the potential for overdiagnosis and resulting overtreatment [2]. Papillary thyroid cancer (PTC) is the most prevalent thyroid cancer of follicular origin, comprising 90% of all thyroid cancers, and it has prolonged survival. Generally, localized or locally advanced PTC can be successfully cured via surgical resection and radioiodine (RAI) ablative treatment. The American Thyroid Association’s current clinical practice guidelines are the leading resources for the diagnosing and treating differentiated thyroid cancer [2]. However, its recurrent and/or metastatic counterparts, especially those that lose their radioiodine accumulation capacity due to dedifferentiation and progression, become untreatable [3]. The five-year survival rate in localized cases stands at 98.1%, but it drastically drops to 55.5% in cases with distant metastases. For patients with RAI-accumulating persistent thyroid cancer, the 10-year survival rate is 29%. Unfortunately, this rate further drops to a mere 10% in cases of lost RAI accumulation [4]. Typically, PTC patients undertake surgery followed by RAI ablation and TSH suppressive therapy as the standard care. Even though repeated radioiodine therapy can slow down the progression of metastatic disease, the tumor inevitably dedifferentiates over time and loses its capacity to accumulate radioiodine. It is well known that the major factors of iodide metabolism in follicular cells of the thyroid are the sodium/iodide symporter (NIS), the thyroid peroxidase (TPO), and thyroglobulin (TG). Radioactive iodine (RAI) is absorbed by thyroid cancer cells through the Na+/I− symporter (NIS). NIS, a plasma membrane glycoprotein, combines the inward movement of sodium (Na+) with the uphill movement of iodine (I−) against its electrochemical gradient. NIS-mediated RAI therapy for thyroid cancer is the pioneering and most successful molecular-targeted radiotherapy available today [5]. If NIS expression is lacking or plasma targeting of NIS is impaired, RAI uptake might be reduced or even absent in thyroid cancer [6]. In one of our recent studies, we first reconfirmed previous observations that NIS expression is absent only in 33% of PTCs, and NIS expression is increased in the majority of thyroid cancer but retained in the cytoplasmatic membrane compartments, resulting in absent or significantly decreased radioiodine accumulation in thyroid cancer cells. The outlook for patients with iodine-resistant thyroid carcinoma is grim, with an average survival period of 3–5 years [7]. The RAI resistance in metastatic PTC, indicated by an absence of RAI accumulation, stems from reduced NIS expression or malfunctioning plasma membrane targeting of the transporter. If we decipher the underlying mechanisms of iodine intake disruption in thyroid cancer, it could facilitate the reinstatement of RAI uptake [8]. Current clinical trials involve the use of mitogen-activated protein kinase (MAPK) inhibitors like selumetinib, dabrafenib, and trametinib as selective BRAF/MEK inhibitors which, by specifically inhibiting the MAPK pathway, restore the experimentation of NIS, which improves radioiodine uptake, as previously described [9]. RAI refractoriness (lack of RAI accumulation) of metastatic PTC is caused by a decrease in NIS expression or impaired plasma membrane targeting of the transporter. Understanding the pathomechanism of iodine uptake disorder in thyroid cancer could open up the possibility of reinduction of RAI uptake. The programmed cell death ligand (PD-L1), an immune checkpoint molecule, is a significant target of FDA (Food and Drug Administration)- and EMA (European Medicines Agency)-approved cancer immunotherapies. PD-L1 expression is being considered a potential biomarker for response of anti-PD-1 or anti-PD-L1 agents in various tumors [10]. PD-L1, expressed on cancer cell surfaces, connects to PD-1 (programmed death 1) on effector T cells, curbing their anticancer effects. In cancer immunotherapy, anti-PD-L1 and/or anti-PD-1 antibodies are used to thwart the PD-L1-PD-1 interaction, ultimately reactivating the anticancer activity of T effector cells [11]. This process, known as “cell-extrinsic” PD-L1–PD-1 interaction, triggers PD-1 downstream signaling that leads to T cell inactivation, and it is well-documented. The newly discovered “cell-intrinsic” PD-L1 signaling, which is largely PD-1-independent, regulates cancer cell proliferation, survival, signaling, and gene expression, among other things [12]. Additionally, anti-PD-L1 antibodies can induce PD-L1-intrinsic signaling when they interact with PD-L1 on the surface of cancer cells. The more thyroid carcinomas dedifferentiate, the more they express the “immune-hijacking” molecule PD-L1 on their surface. As a result, tumor cells become RAI refractory during further dedifferentiation and eventually stop taking up RAI. Thus, thyroid cancer dedifferentiation often results in high PD-L1 expression and no RAI uptake.

As our study is the first attempt to explore possible association between PD-L1 and NIS expression, it has several limitations.

## 2. Materials and Methods

Patients were operated upon at the Head and Neck Surgical Department in the National Institute of Oncology, Budapest, Hungary due to metastatic papillary thyroid cancer. A retrospective immunohistochemistry examination of paraffin-embedded samples from 89 randomly chosen patients with regional lymph node metastatic papillary thyroid cancer was performed. All the formalin-fixed, paraffin-embedded tissue block samples originated from the archives of the Center of Tumor Pathology, National Institute of Oncology. These patients were operated upon by high-volume thyroid surgeons performing thyroidectomy with central and with/without lateral neck lymph node dissection at the same institution between 1 January 2013 and 31 December 2018. All data were fully anonymized. In both PD-L1 and NIS staining, the procedures were the same, and the reactions were performed on sections cut from the same tissue block. The fresh tissue from the patients was fixed in 10% neutral buffered formalin for a minimum of 12 h and embedded in paraffin. The formalin-fixed paraffin-embedded (FFPE) tissue was then cut to 5-micron thin sections and treated as per the manufacturer’s guidelines using the anti-PD-L1 Dako22C3 PharmDx kit(The Dako PD-L1 22C3 pharmDx™ kit, KEYTRUDA^®^ (pembrolizumab), Rahway, NJ, USA). The tissue slide was heated to 72 Celsius for 4 min of incubation time for the deparaffination, followed by heating to 95 Celsius for 8 min in order to achieve cell pre-conditioning with Ultra Conditioner, and for another 36 min with Ultra CC1 Conditioner for further cell conditioning. After these steps, the primary antibody was added to the section with an incubation time of 40 min. The procedure ended with hematoxylin background staining. The hNIS expression was examined using immunohistochemistry, employing a rabbit polyclonal antibody against the C terminal end of hNIS at a dilution of 1:4000 [8,9], a generous contribution from Dr. Nancy Carrasco. Immunohistochemistry was performed as previously mentioned [10,11], with all slides subsequently counterstained with hematoxylin. Stains from the immunohistochemical process were assessed and scored by two experienced pathologists specializing in PD-L1 and NIS staining, using both individual and multi-headed microscopes (Nikon Eclipse E200). The samples were examined by endocrine pathologists who are experts in thyroid pathology. As the variables did not show Gaussian distribution (using the Shapiro–Wilk test), a non-parametrical Spearman rank order correlation test was utilized to ascertain the relationship between PD-L1 (TPS) and NIS expression in PTC tumors. (Statistica 12.5, StatSoft, Tulsa, OK, USA). The significance level was 0.05. Analysis of programmed death ligand 1 (PD-L1) and sodium iodide symporter (NIS) expression in papillary thyroid cancer (PTC) sections detected by immunohistochemistry (IHC).

Tumor cells were grouped according to the percentage showing PD-L1 staining, measured by the tumor proportion score (TPS). There were three groups: the first had no staining (score 0); the second had between 1% and 50% staining (score 1); the third had more than 50% staining (score 2). After assessing PD-L1 staining, NIS immunohistochemistry analysis was performed on 85 cases, and the percentage of tumor cells showing cytoplasmic and/or plasma membrane NIS expression was documented. We had also interest in investigating and correlating PD-L1 and NIS expression in lymph node metastatic papillary thyroid cancer samples and correlating these with the size, multifocality, metastatic nature, and iodide-accumulating ability (i.e., the plasma membrane expression of NIS) of the differentiated thyroid cancer.

## 3. Results

Immunohistochemistry analysis of PD-L1 was conducted on the primary tumor from 89 regional lymph node metastatic PTC cases. Of these tumors, 25 (28%) showed no PD-L1 expression. Meanwhile, 58 (65%) and 6 (7%) of the tumor tissues fell into staining groups 1 and 2, respectively.

NIS immunohistochemistry was conducted on 86 primary papillary carcinomas. Out of these, 51 (60%) tumors exhibited NIS expression. Notably, only seven (8%) cases demonstrated NIS localization in the plasma membrane, while in the majority of tumors, NIS was retained within intracytoplasmic membrane compartments (Figure 1, Table 1). NIS and PD-L1 expression were evaluated on separate tissue sections from the same tumor. Unfortunately, we were unable to investigate the co-expression of NIS and PD-L1 in tumor cells. We found that PD-L1 (TPS) has no correlation with the number of metastatic lymph nodes (*p* = 0.1323) or with capsular invasion (*p* = 0.0937), or with lymphatic invasion (*p* = 0.3123). On the other hand, significant correlation was detected with vascular invasion (*p* = 0.0165) (Figure 2). Furthermore we found no correlation between NIS and the number of metastatc lymph nodes (*p* = 0.6786), with capsular invasion (*p* = 0.9567), or with lymphatic invasion (*p* = 0.7673). There was also no correlation found with vascular invasion (*p* = 0.5945). Regarding the correlation of PD-L1 and NIS expression with age, tumor size and multifocality, we found a strong correlation between the TPS expression and (0.0183, R = 0.24) and the tumor size (*p* = 0.0237, R = 0.24), but no correlation was detected with multifocality. NIS did not show any correlation with the multifocality (*p* = 0.5182) or tumor size (*p* = 0.7784). Notably, there was no apparent correlation between PD-L1 and NIS expression (*p* = 0.3199).

The NIS value in the table was calculated by the percentage of cells showing staining with any intensity of any subcellular compartment (cytoplasmic and/or plasma membrane). Using the aforementioned calculation, we can complete the column of NIS as follows: ‘NIS total (cytoplasmic + plasma membrane)’. The Spearman rank order correlation test was utilized to ascertain the relationship between PD-L1 (TPS) and NIS expression in PTC tumors. Notably, there was no apparent correlation between PD-L1 and NIS expression (*p* = 0.3199). No correlation was found between PD-L1 expression (TPS) and metastatic lymph nodes (Table 2).

## 4. Discussion

Thyroid cancer currently ranks as the 13th most common cancer diagnosis overall and the 6th most common among women. Its incidence is increasing worldwide, while its mortality rate has been slightly increasing in recent years. Thyroid malignancies have a 3:1 female predominance. In our current study, we reaffirmed earlier observations, noting an absence of NIS expression in only 33% of papillary thyroid cancers [6,13]. Untreated thyroid cancer can be locally aggressive and may invade the recurrent laryngeal nerve(s), airway, esophagus, or other nearby neurovascular structures. Distant metastasis most commonly involves the lung or bone. We also found that despite NIS expression increasing in most thyroid cancers, it remains in the cytoplasmic membrane compartments. This results in a significant reduction, or even absence, of radioiodine accumulation in thyroid cancer cells. Others reported that papillary thyroid cancer with a thyroiditis background demonstrated much higher PD-L1 expression compared to papillary thyroid cancer with a normal background [14]. In our study we did not evaluate the presence of thyroiditis and its association with PD-L1 or NIS expression.

The thyroid gland is the most frequently involved in cancer and autoimmunity. In addition, the risk of developing thyroid cancer is increased in patients with autoimmune thyroid disease (e.g., Hashimoto thyroiditis or Grave’s disease), and in these patients, increased levels of circulating and intrathyroidal PD-1-positive T-cells, among others, can be detected. Additionally, in the line with results of others, we also demonstrated that a significant proportion (72%) of PTSs express PD-L1.

Several studies have demonstrated that during the dedifferentiation process in most thyroid cancer, the expression of PD-L1 increases, and NIS functional expression decreases. The PD-1/PDL-1 pathway in general is used by cancer cells to resist immune destruction, while a lack of functional NIS expression leads to radioiodine resistance [15].

PD-L1 expression was evaluated only in the cancer cells and expressed as TPS (tumor progression score); the PD-L1 staining of tumor infiltrating immunocells was not recorded.

We also did not investigate PD-L1 or NIS expression in corresponding lymph node metastases, but others have reported that PD-L1 expression in metastatic papillary thyroid cancer tissues is similar to their corresponding primary tumor in the thyroid [16].

Our results align with other findings demonstrating that a significant 72% of papillary thyroid cancers express PD-L1 [14]. However, we found no correlation between NIS and PD-L1 expression in the primary tumors of papillary thyroid cancer with lymph node metastases.

Lubin et al. revealed that PTC with a background of Hashimoto thyroiditis significantly expresses PD-L1 [17]. Other studies have also shown that PTCs with a thyroiditis background express much more PD-L1 compared to those with a normal background [18,19]. However, our study did not explore the potential association between background thyroiditis and either PD-L1 or NIS expression.

Unfortunately, the evaluation of the association between NIS and PD-L1 in the same cancer cell was not possible, as we were unable to conduct co-localization studies of NIS and PD-L1 expression at the cellular level. Numerous studies illustrate that in most thyroid cancers, PD-L1 expression increases, and NIS functional expression decreases during the dedifferentiation process [3,20]. The PD-1/PDL-1 pathway generally aids cancer cells in evading immune destruction. Concurrently, the absence of functional NIS expression results in resistance to radioiodine.

These two processes may be related, but they could also be independent outcomes of the dedifferentiation process. The authors pointed out that NOX4 could serve as a potential therapeutic target along with other MAPK-kinase inhibitors aiming to potentiate their effect on RAI-refractory DTC redifferentiation.

## 5. Conclusions

In conclusion, we did not find any correlation between the percentage of NIS- and PD-L1-expressing tumor cells in the primary tumors of lymph node metastatic PTCs.

It is intriguing to consider whether the use of anti-PDL1 or anti-PD1 antibodies could rejuvenate NIS functional expression by manipulating the PD-1/PD-L1 axis. However, based on the current study, we can only conclude that there is no correlation between the percentage of NIS-expressing and PD-L1-expressing tumor cells in the primary tumor of lymph node metastatic papillary thyroid cancer.

## Figures and Tables

**Figure 1 diagnostics-14-01858-f001:**
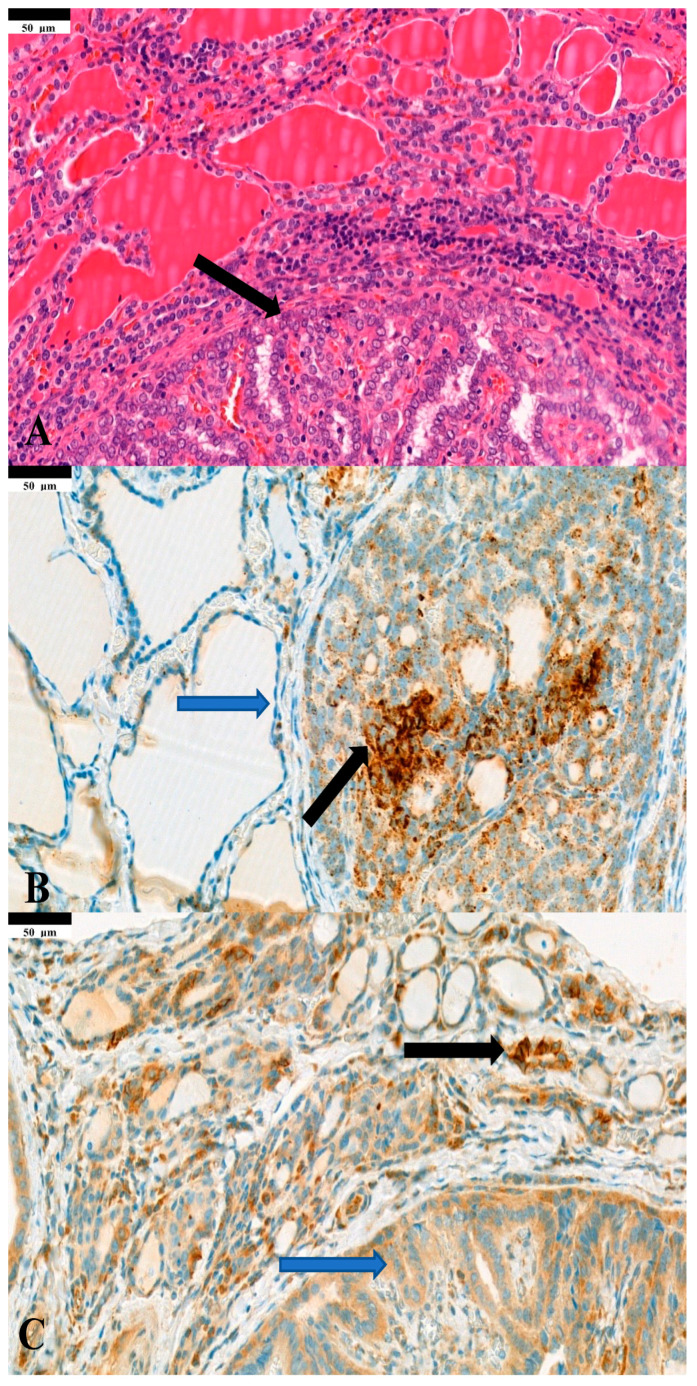
(**A**) Papillary thyroid carcinoma showing typical architectural and cytomorphological features (black arrow) (HE: hematoxylin–eosin staining). (**B**) Papillary thyroid carcinoma cells featuring membranous PD-L1 staining (black arrow) and normal follicular cells showing no staining (blue arrow) (PD-L1 IHC: immunohistochemistry). (**C**) Normal follicular epithelial cells exhibiting heterogenous NIS staining on their membrane (black arrow) compared with tumor cells exhibiting intracellular NIS. (**A**) HE staining; (**B**) PD-L1 IHC; (**C**) NIS IHC. (40× magnification; all samples were taken from the same tumor sample).

**Figure 2 diagnostics-14-01858-f002:**
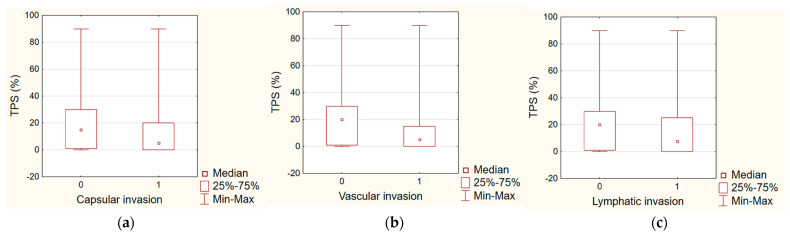
PD-L1 expression (TPS) depending on capsular (**a**), vascular (**b**), and lymphatic invasion (**c**).

**Table 1 diagnostics-14-01858-t001:** IHC expression of PD-L1 and NIS in PTC. PD-L1: programmed death ligand 1, range: 1 = 0%, 2 = 1–50%, 3 = 51–100%, NIS: sodium iodide symporter.

Parameters	Mean (Min–Max)
PD-L1 (%)	17.7 (0.0–90.0)
Range	1.8 (1.0–3.0)
NIS total (%)	18.7 (0.0–100.0)
NIS only citoplasmatic (%)	17.2 (0.0–100.0)
NIS plasmamembrane (%) also present	1.5 (0.0–30.0)

**Table 2 diagnostics-14-01858-t002:** Relationship (*p* values) of PD-L1 and NIS staining with the clinicopathological data of the patients (NS: non-significant).

Clinicopathological Parameter	PD-L1	NIS
Gender	NS	NS
Age	0.0183	NS
TNM stage	NS	NS
Tumor size	0.0237	NS
Multifocality	NS	NS
(Nr.) Number of metastatic lymph nodes	NS	NS
Capsular invasion	NS	NS
Vascular invasion	0.0165	NS
Lymphatic invasion	NS	NS

## Data Availability

All data used in the preparation of the paper are available on reasonable request.

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
