# Peer review of "No Correlation between PD-L1 and NIS Expression in Lymph Node Metastatic Papillary Thyroid Carcinoma"

_diagnostics, 2024, doi:10.3390/diagnostics14171858_

Round 1

Reviewer 1 Report

Comments and Suggestions for Authors

comments to the manuscript ID 3094335, where the authors evaluate the possible association between PD-L1 and NIS expression in Lymph Node Metastatic Papillary Thyroid Carcinoma. Some comments are listed below.

1 respect the scores and spaces after the dots, comas and references

2 the introduction is extensive compared to the results described, it is suggested to shorten it

3 in the statistics part, indicate the statistical proof of the relationship analysis, delve deeper into the statistics used

4 improve the quality of the images presented and the arrows make them narrower

5 include a brief explanation and title of figures 2, 3, and 4

6 Table 1 defines the units (n), (%)? , present the table as supplementary material

6 conclusion leave only the first two lines, including the rest at the end of the discussion, ensuring that the text is not repeated. You can improve your discussion based on the results found, and propose theories.

Author Response

Dear Reviewer 1,

thanks you very much for the quick and appropriate comments.

We give answers and comments below :

1 respect the scores and spaces after the dots, comas and references

1. We corrected the text and the references as well.

2 the introduction is extensive compared to the results described, it is suggested to shorten it

2. We shortened the introduction as needed

3 in the statistics part, indicate the statistical proof of the relationship analysis, delve deeper into the statistics used

3. We corrected the statistical part:

As the variables did not shown Gaussian distribution (using Shapiro-Wilks test), non-parametrical Sperman-Rank Order Correlation was run to determine the relationship between PD-L1 (TPS) and NIS expression in PTC tumors (Statistica 12.5, StatSoft, Tulsa, OK, USA). The significance level was 0.05.

4 improve the quality of the images presented and the arrows make them narrower

4.We improved the quality of the images presented, made the arrows narrower.

5 include a brief explanation and title of figures 2, 3, and 4

5.We corrected  and explained the Figures 2,3,4 and made one figure as Reviewer 2.

Figure 2. PD-L1 expression (TPS) did not shown correlation with a, capsular (p=0.0937) and c, lymphatic invasion, however, correlated with b, vascular invasion (p=0.0165). Furthermore we found no correlation between the total NIS expression and the number of metastatic lymph nodes.

6 Table 1 defines the units (n), (%)? , present the table as supplementary material

6. We designed a new Table 1. as reviewer 2 and 3 commented with the units.

Table 1. IHC expression of PD-L1 and NIS in PTC. PD-L1:programmed death ligand 1,  Range: 1=0%, 2=1-50%, 3= 51-100%, NIS: natrium-iodide symporter

PD-L1 (%)

17.7 (0.0-90.0)

Range

1.8 (1.0-3.0)

NIS total (%)

18.7 (0.0-100.0)

NIS only citoplasmatic (%)

17.2 (0.0-100.0)

NIS

plasmamembrane (%) also present

1.5 (0.0-30.0)

7 conclusion leave only the first two lines, including the rest at the end of the discussion, ensuring that the text is not repeated. You can improve your discussion based on the results found, and propose theories.

7. We shortened the conclusion as requested.

Reviewer 2 Report

Comments and Suggestions for Authors

The manuscript “No correlation between PD-L1 and NIS expression in lymph node metastatic papillary thyroid carcinoma”, diagnostics-3094335, presents the IHC staining of PD-L1 and NIS in PTC patients with lymph node metastasis in order to elucidate if there is any correlation between their expression in the tested sample. Unfortunately, the study presented in this way contains many flaws, many of which are major. 

1.        Abstract:

a.        There is no explanation of the PD-L1 acronym (programmed death ligand-1)

b.       The IHC staining of both markers should be presented in a uniform way. For both markers: distribution of staining, subcellular localization and the cut off value for positivity should be clearly presented. In this version of the abstract, the authors showed a distribution of PD-L1 IHC staining in % with no subcellular localization, while staining of NIS was presented via positive and negative IHC staining in subcellular compartments.  

c.        The last line of the abstract should be rewritten as written in this way; it does not present the main conclusion correctly. Therefore, instead of: “However, based on our study, the only conclusion that can be drawn is that there is not a correlation between the percentage of either NIS or PD-L1 expressing tumor cells in the primary tumor of lymph node metastatic PTC”, I suggest you simplify: “…. There is no correlation between NIS and PD-L1 expression in PTC…”

2.        The introduction section:

a.        It is too long; it should be shortened.

b.       All statements written between lines 47 and 63 need to be referenced.

c.        The limitations of the study should be transferred into the discussion section of the manuscript.

d.       The acronym “TPS” (line 141) needs an explanation.

3.        Material and methods section:

a.        Line 153: it seems that the data were partly, or pseudo anonymized, not fully. Please correct

b.       Please name the scoring of NIS staining (group categories). Was it the same as PD-L1?

c.        Name the cutoff value for the positive IHC result, for both markers.

d.       Name the statistical tests applied (which test was applied for which comparison).

4.        Results – In general, the presentation of the results should be significantly improved:

a.        Table 1 should be transformed. The IHC staining results should not be presented via staining of each sample, but the cases should be grouped in a way the authors choose. You may show your results via distribution, median or average values, percentage, number of cases in each categories.

i.        Each symbol/word/acronym used in the table (including the table’s title) must be described under that table

ii.        The table title must reflect the table content, including sample type and methodology (e.g, IHC expression of PD-L1 and NIS in PTC)

iii.       For each parameter in the table, the metric should be named (e.g. %)

iv.        Describe 1,2,3 values of range under the table. Is it the intensity of the staining?

 v.       As I could understand, NIS value in the table is calculated as NIS cyt + NIS mem. If so, what about the cases that showed positive both, cytoplasmic and membrane staining of the same cells (e.g. the case presented in Figure 1)? Total staining in cases like that one could not be calculated as a simple summation of membrane and cytoplasmic %, as the real % of positive cells is less due to the superposition of staining.

b.       Line 188: Please, rewrite the line “We examined PD-L1 (TPS) no correlation with the number of metastatic lymph nodes (p=0,1323) neither with capsular invasion (p=0,0937).” , as written in this way, the sentence is incomplete, incorrect (by English grammar) and unclear. In general, all sentences written between lines 188 and 198 need extensive English editing.

c.        I suggest you present the results of the correlation of PD-L1 and NIS IHC staining with the clinicopathological data of the patients in a new separate table

d.       Figure 1: the right part of the figure (the textual part) should be removed as it is all explained under the figure.

e.        Figures 2, 3, and 4 should be merged and presented via one figure with 3 graphs marked with a, b, c marks. Furthermore, texts above the graphs should be deleted and y-axes must be defined (TPS %)

f.          Line 214: “The Spearman-Rank Order Correlation test was utilized to ascertain the relationship between PD-L1 (TPS) and NIS expression in PTC tumors”. The values from which two columns in Table 1 were correlated?

g.        Line 217: “No correlation was found between PD-L1 expression (TPS) and metastatic lymph nodes.” This sentence is not clear. What did you want to say with this line? Did you correlate TPS with the occurrence of lymph node metastasis in PTC patients?

5.        Discussion: the discussion section is very poor and short. It must be significantly improved, more previously published data needs to be mentioned, described and discussed. Particularly if there are some divergent results.

6.        The conclusion needs to be rewritten. It should be more focused and not presented in the same way as the discussion section.

7.        There are two sentences that are almost the same. In line 246: “Numerous studies illustrate that in most thyroid cancers, PD-L1 expression increases, and NIS functional expression decreases during the dedifferentiation process” and in line 250: “In most thyroid cancer cases, PD-L1 expression increases while functional NIS expression decreases during dedifferentiation”. Please delete one.

Comments on the Quality of English Language

Some parts of the manuscript should be edited/rewritten

Author Response

  1. Abstract:
  2. There is no explanation of the PD-L1 acronym (programmed death ligand-1) We explained
  3. The IHC staining of both markers should be presented in a uniform way. For both markers: distribution of staining, subcellular localization and the cut off value for positivity should be clearly presented. In this version of the abstract, the authors showed a distribution of PD-L1 IHC staining in % with no subcellular localization, while staining of NIS was presented via positive and negative IHC staining in subcellular compartments. We corrected as the following: with NIS IHC any subcellular staining was counted as positive
  4. The last line of the abstract should be rewritten as written in this way; it does not present the main conclusion correctly. Therefore, instead of: “However, based on our study, the only conclusion that can be drawn is that there is not a correlation between the percentage of either NIS or PD-L1 expressing tumor cells in the primary tumor of lymph node metastatic PTC”, I suggest you simplify: “…. There is no correlation between NIS and PD-L1 expression in PTC…”We corrected as requested:There was no observed correlation between PD-L1 and NIS expression.
  5. The introduction section:
  6. It is too long; it should be shortened. Shortened as requested.
  7.      All statements written between lines 47 and 63 need to be referenced. References were included
  8. The limitations of the study should be transferred into the discussion section of the manuscript. The limitations were transferred into the discussion part.
  9. The acronym “TPS” (line 141) needs an explanation. We explained.
  10. Material and methods section:
  11. Line 153: it seems that the data were partly, or pseudo anonymized, not fully. Please correct Data were fully anonymized.
  12. Please name the scoring of NIS staining (group categories). Was it the same as PD-L1? ,  Range: 1=0%, 2=1-50%, 3= 51-100%, 
  13. Name the cutoff value for the positive IHC result, for both markers.

    The NIS value in the table is calculated by the percentage of cells showing staining with any intensity of any subcellular compartment (citoplasmic and/or plasmamembrane). According to the aforementioned we complete the column of NIS as follows: 'NIS total (citoplasmic + plasmamembrane)'

    The next column of the table will be modified as 'NIS only citoplasmic' and the last column will be modified as 'NIS plasmamembrane also present' as the last column was made to show the fraction of tumor cells showing also plasmamembrane positivity with or without citoplasmic staining. We think that this modified form of presentaion will be straitforward in scope of calculation.

  14. Name the statistical tests applied (which test was applied for which comparison).The Spearman-Rank Order 
  15. Results – In general, the presentation of the results should be significantly improved:We improved the result section
  16. Table 1 should be transformed. The IHC staining results should not be presented via staining of each sample, but the cases should be grouped in a way the authors choose. You may show your results via distribution, median or average values, percentage, number of cases in each categories. Table 1 was modified as requested.
  17. Each symbol/word/acronym used in the table (including the table’s title) must be described under that table They were described as requested
  18. The table title must reflect the table content, including sample type and methodology (e.g, IHC expression of PD-L1 and NIS in PTC) We modified the table title

iii.       For each parameter in the table, the metric should be named (e.g. %) Metrics were named

  1. Describe 1,2,3 values of range under the table. Is it the intensity of the staining?1=0%, 2=1-50%, 3= 51-100%, 
  2.  As I could understand, NIS value in the table is calculated as NIS cyt + NIS mem. If so, what about the cases that showed positive both, cytoplasmic and membrane staining of the same cells (e.g. the case presented in Figure 1)? Total staining in cases like that one could not be calculated as a simple summation of membrane and cytoplasmic %, as the real % of positive cells is less due to the superposition of staining. The NIS value in the table is calculated by the percentage of cells showing staining with any intensity of any subcellular compartment (citoplasmic and/or plasmamembrane). According to the aforementioned we complete the column of NIS as follows: 'NIS total (citoplasmic + plasmamembrane)' The next column of the table will be modified as 'NIS only citoplasmic' and the last column will be modified as 'NIS plasmamembrane also present' as the last column was made to show the fraction of tumor cells showing also plasmamembrane positivity with or without citoplasmic staining. We think that this modified form of presentaion will be straitforward in scope of calculation.
  3. Line 188: Please, rewrite the line “We examined PD-L1 (TPS) no correlation with the number of metastatic lymph nodes (p=0,1323) neither with capsular invasion (p=0,0937).” , as written in this way, the sentence is incomplete, incorrect (by English grammar) and unclear. In general, all sentences written between lines 188 and 198 need extensive English editing.We corrected the English.
  4. I suggest you present the results of the correlation of PD-L1 and NIS IHC staining with the clinicopathological data of the patients in a new separate table Clinicopathological data were described in a separate table as requested 

    Table 2. Correlations (p values) of PD-L1 and NIS staining with the clinicopathological data of the patients (NS: non-significant).

    Clinicopathological parameter

    PD-L1

    NIS

    Gender

    NS

    NS

    Age

    0.0183

    NS

    TNM stage

    NS

    NS

    Tumour size

    0.0237

    NS

    Multifocality

    NS

    NS

    Nr. of metastatic lymph nodes

    NS

    NS

    Capsular invasion

    NS

    NS

    Vascular invasion

    0.0165

    NS

    Lymphatic invasion

    NS

    NS

  5. Figure 1: the right part of the figure (the textual part) should be removed as it is all explained under the figure. The right part is removed.
  6. Figures 2, 3, and 4 should be merged and presented via one figure with 3 graphs marked with a, b, c marks. Furthermore, texts above the graphs should be deleted and y-axes must be defined (TPS %) One figure is made from 2,3,4 Figures and corrected per request.
  7. Line 214: “The Spearman-Rank Order Correlation test was utilized to ascertain the relationship between PD-L1 (TPS) and NIS expression in PTC tumors”. The values from which two columns in Table 1 were correlated? Table 1. is corrected .
  8. Table 1. IHC expression of PD-L1 and NIS in PTC. PD-L1:programmed death ligand 1,  Range: 1=0%, 2=1-50%, 3= 51-100%, NIS: natrium-iodide symporter

    PD-L1 (%)

    17.7 (0.0-90.0)

    Range

    1.8 (1.0-3.0)

    NIS total (%)

    18.7 (0.0-100.0)

    NIS only citoplasmatic (%)

    17.2 (0.0-100.0)

    NIS

    plasmamembrane (%) also present

    1.5 (0.0-30.0)

  9. Line 217: “No correlation was found between PD-L1 expression (TPS) and metastatic lymph nodes.” This sentence is not clear. What did you want to say with this line? Did you correlate TPS with the occurrence of lymph node metastasis in PTC patients? Yes, with the occurrence, corrected in the text.
  10. Discussion: the discussion section is very poor and short. It must be significantly improved, more previously published data needs to be mentioned, described and discussed. Particularly if there are some divergent results. Corrected.
  11. The conclusion needs to be rewritten. It should be more focused and not presented in the same way as the discussion section.Corrected.
  12. There are two sentences that are almost the same. In line 246: “Numerous studies illustrate that in most thyroid cancers, PD-L1 expression increases, and NIS functional expression decreases during the dedifferentiation process” and in line 250: “In most thyroid cancer cases, PD-L1 expression increases while functional NIS expression decreases during dedifferentiation”. Please delete one. Deleted the first.

Comments on the Quality of English Language

Some parts of the manuscript should be edited/rewritten

Text was edited and corrected by the American Manuscript Editor system.

Reviewer 3 Report

Comments and Suggestions for Authors

The manuscript by Bernadett and co-workers focuses on the connection between PD-L1 and NIS expression in lymph node metastatic papillary thyroid carcinoma (PTC). Unfortunately, I do not recommend the article in its current form for publication. The manuscript requires very significant changes, as suggested below.

[1] The Introduction is too long, especially compared to the very concise Discussion. It is worth moving some sections of the Introduction to the Discussion to improve this ratio. Moreover, the limitations of the study are described twice - once in the Introduction and the second time in the Discussion. The description of the study's limitations should appear only in the Discussion.

[2] In some part of the manuscript there is no reference to the publication from which the information provided is taken, e.g., in lines 87, 118 and 145.

[3] It is worthwhile to refer to more recent work describing the radioiodine refractory (RAIR) phenomenon in thyroid carcinoma, e.g., Endocr Relat Cancer. 2022 Apr 22;29(5):R57-R66. doi: 10.1530/ERC-22-0006.

[4] There is a lack of description of the patients (age, gender, TNM scale, etc.), from whom the tissue sections were obtained. It is worth presenting these data in the form of a table. In the Materials and Methods section, the Authors state that “all patients provided informed written consent to have data from their medical records used in research”. On the other hand, in the Informed Consent Statement section, the Authors write that “Patient consent was waived due to that patients cannot be identified.” These contradictory statements require clarification.

[5] Figure 1: It is worth noting that the images presented are representative. A scale should be added in each image. Additionally, it is worth introducing insets in which there will be a magnification of a section of tissue, where the subcellular localization of the proteins under study will be better visible. Explain: IHC and HE.

[6] Table 1: The legend does not sufficiently describe the data presented in the table. For example, it is not clear what the numbers in the table are, what the “Range” header means. Some rows are missing a complete set of data (Patient No. 54 and 70) - this needs clarification.

[7] It is not explained why only some of the data is presented graphically (Figs. 2-4), and some is only briefly described in lines 214-218. Legends should be added to Figs. 2-4. Moreover, the font size in Figs. 2-4 should be increased and statistical analysis should be added.

[8] In Materials and Methods, a separate subsection should be added describing the statistical analysis in detail, including the computer program for statistical analysis, statistical test used to check normality of distribution, the p-value that was found to be borderline (p < 0.05?), etc.

[9] Line 162: The description of immunohistochemistry is definitely incomplete. Please at least briefly describe all the steps from the excision of the tissue and sequentially the steps including processing, embedding, cutting and immunodetection. When describing the method, the Authors provide a reference [10], which is a meta-analysis. I am not sure if the meta-analysis actually includes a detailed description of PD-L1 and NIS detection in tissue sections. If known, please provide the exact amino acid sequence recognized by the anti-NIS antibody (in line 160). Finally, please provide details about the microscope used in the study?

[10] Lines 191-193: Please clarify whether the Authors are referring to the expression of total NIS, or the intracellular fraction, or the fraction embedded in the plasma membrane.

[11] Line 110: Please explain what “PTSs” (line 110), “EMA” and “FDA“ (line 103) stand for. The explanation of the abbreviation “TPS” should appear only once in the first use in the text, that is, in line 141 (and now it is explained only in line 168).

[12] In the Introduction, the Authors only mention NIS symporter as the only protein that is important in the accumulation of iodine-131 radioisotope in thyrocytes. However, it is worth noting the role of proteins involved in iodine organization, e.g., thyroid peroxidase (TPO), whose expression also declines in thyroid carcinoma (as described in Biochimie. 2019 May;160:34-45. doi: 10.1016/j.biochi.2019.02.003). It seems that iodine organization prolongs its retention in thyrocytes, which increases the chances of destroying cancer foci.

[13] In lines 89-90, it is worth mentioning selective BRAF/MEK inhibitors (especially dabrafenib and trametinib), which, by specifically inhibiting the MAPK pathway, restore the experimentation of NIS, which improves radioiodine uptake, as previously described (Int J Mol Sci. 2021 Oct 31;22(21):11829. doi: 10.3390/ijms222111829).

Examples of typos:

[1] Lines 113, 115 and 117: Change “PD1” for “PD-1”.

[2] Table 1: Correct “Case N” and “plasmamembrane”. 

Author Response

Dear Reviewer 3,

Thank you very much for the quick and appropriate comments.

The Introduction is too long, especially compared to the very concise Discussion. It is worth moving some sections of the Introduction to the Discussion to improve this ratio. Moreover, the limitations of the study are described twice - once in the Introduction and the second time in the Discussion. The description of the study's limitations should appear only in the Discussion.

Corrected as requested.

[2] In some part of the manuscript there is no reference to the publication from which the information provided is taken, e.g., in lines 87, 118 and 145.

References were added.

[3] It is worthwhile to refer to more recent work describing the radioiodine refractory (RAIR) phenomenon in thyroid carcinoma, e.g., Endocr Relat Cancer. 2022 Apr 22;29(5):R57-R66. doi: 10.1530/ERC-22-0006.

Reference added.

[4] There is a lack of description of the patients (age, gender, TNM scale, etc.), from whom the tissue sections were obtained. It is worth presenting these data in the form of a table. In the Materials and Methods section, the Authors state that “all patients provided informed written consent to have data from their medical records used in research”. On the other hand, in the Informed Consent Statement section, the Authors write that “Patient consent was waived due to that patients cannot be identified.” These contradictory statements require clarification.

Table 2. describes the clinicopathological data.

Method section was clarified.

[5] Figure 1: It is worth noting that the images presented are representative. A scale should be added in each image. Additionally, it is worth introducing insets in which there will be a magnification of a section of tissue, where the subcellular localization of the proteins under study will be better visible. Explain: IHC and HE.

Figure 1 was corrected as requested. Magnification and scales were added.

[6] Table 1: The legend does not sufficiently describe the data presented in the table. For example, it is not clear what the numbers in the table are, what the “Range” header means. Some rows are missing a complete set of data (Patient No. 54 and 70) - this needs clarification.

Table one corrected and modified.

Table 1. IHC expression of PD-L1 and NIS in PTC. PD-L1:programmed death ligand 1,  Range: 1=0%, 2=1-50%, 3= 51-100%, NIS: natrium-iodide symporter

PD-L1 (%)

17.7 (0.0-90.0)

Range

1.8 (1.0-3.0)

NIS total (%)

18.7 (0.0-100.0)

NIS only citoplasmatic (%)

17.2 (0.0-100.0)

NIS

plasmamembrane (%) also present

1.5 (0.0-30.0)

[7] It is not explained why only some of the data is presented graphically (Figs. 2-4), and some is only briefly described in lines 214-218. Legends should be added to Figs. 2-4. Moreover, the font size in Figs. 2-4 should be increased and statistical analysis should be added.

Figures corrected and Statistical analysis added.

As the variables did not shown Gaussian distribution (using Shapiro-Wilks test), non-parametrical Sperman-Rank Order Correlation was run to determine the relationship between PD-L1 (TPS) and NIS expression in PTC tumors (Statistica 12.5, StatSoft, Tulsa, OK, USA). The significance level was 0.05.

[8] In Materials and Methods, a separate subsection should be added describing the statistical analysis in detail, including the computer program for statistical analysis, statistical test used to check normality of distribution, the p-value that was found to be borderline (p < 0.05?), etc.

As the variables did not shown Gaussian distribution (using Shapiro-Wilks test), non-parametrical Sperman-Rank Order Correlation was run to determine the relationship between PD-L1 (TPS) and NIS expression in PTC tumors (Statistica 12.5, StatSoft, Tulsa, OK, USA). The significance level was 0.05.

[9] Line 162: The description of immunohistochemistry is definitely incomplete. Please at least briefly describe all the steps from the excision of the tissue and sequentially the steps including processing, embedding, cutting and immunodetection. When describing the method, the Authors provide a reference [10], which is a meta-analysis. I am not sure if the meta-analysis actually includes a detailed description of PD-L1 and NIS detection in tissue sections. If known, please provide the exact amino acid sequence recognized by the anti-NIS antibody (in line 160). Finally, please provide details about the microscope used in the study?

In case of both PD-L1 and NIS staining the procedures were the same and the reactions were made on sections cut from the same tissue block. The fresh tissue from the patients was fixed in 10 % neutral buffered formalin for a minimum of 12 hours and embedded in paraffin. The formalin fixed paraffin embedded (FFPE) tissue was than cut to 5 micron thin sections and treated as per the manufacturer's guidelines using the anti-PD-L1 Dako22C3 PharmDx kit. The tissue slide is heated to 72 Celsius for 4 minutes of incubation time for the deparaffination followed by heating to 95 Celsius for 8 minutes in order to cell pre-conditioning with Ultra Conditioner and for another 36 minutes with Ultra CC1 Conditioner for further cell conditioning. After these steps the primary antibody is added to the section with an incubation time of 40 minutes. The procedure is ended with hematoxylin background staining.

[10] Lines 191-193: Please clarify whether the Authors are referring to the expression of total NIS, or the intracellular fraction, or the fraction embedded in the plasma membrane. Total NIS

[11] Line 110: Please explain what “PTSs” (line 110), “EMA” and “FDA“ (line 103) stand for. The explanation of the abbreviation “TPS” should appear only once in the first use in the text, that is, in line 141 (and now it is explained only in line 168). Abbreviations were explained.

[12] In the Introduction, the Authors only mention NIS symporter as the only protein that is important in the accumulation of iodine-131 radioisotope in thyrocytes. However, it is worth noting the role of proteins involved in iodine organization, e.g., thyroid peroxidase (TPO), whose expression also declines in thyroid carcinoma (as described in Biochimie. 2019 May;160:34-45. doi: 10.1016/j.biochi.2019.02.003). It seems that iodine organization prolongs its retention in thyrocytes, which increases the chances of destroying cancer foci.

Reference added.

[13] In lines 89-90, it is worth mentioning selective BRAF/MEK inhibitors (especially dabrafenib and trametinib), which, by specifically inhibiting the MAPK pathway, restore the experimentation of NIS, which improves radioiodine uptake, as previously described (Int J Mol Sci. 2021 Oct 31;22(21):11829. doi: 10.3390/ijms222111829).

Reference added.

Examples of typos:

[1] Lines 113, 115 and 117: Change “PD1” for “PD-1”. Corrected

[2] Table 1: Correct “Case N” and “plasmamembrane”.  Corrected

Round 2

Reviewer 2 Report

Comments and Suggestions for Authors

Dear authors,

The revised version of your manuscript entitled “No Correlation between PD-L1 and NIS Expression in Lymph Node Metastatic Papillary Thyroid Carcinoma” is insufficiently improved. Before of all, many of the previous remarks are not introduced into the revised version (although the authors said that they have introduced them) and there are repeated sentences/lines: 

1.      The IHC staining of both markers is not presented in a uniform way, as requested.

2.      The authors said that the sentence “However, based on our study, the only conclusion that can be drawn is that there is not a correlation between the percentage of either NIS or PD-L1 expressing tumor cells in the primary tumor of lymph node metastatic PTC”, has been simplified, as requested. But they did not do it. There is still the line ‘However, based on our study, the only conclusion that can be drawn is that there is not a correlation between the percentage of either NIS or PD-L1 expressing tumor cells in the primary tumor of lymph node metastatic PTC’ in the abstract. In addition, this sentence is not clear, and it is insufficient. Please, delete it.

3.      The authors did not reference the lines (between 47 and 63 lines in v1, and 62-73 in v2), as requested (although they said that they did it)

4.      Line 194-195 ‘Notably, there was no apparent correlation between PD-L1 and NIS expression (p=0.3199)’ should be transferred into the results section.

5.      There are two Table 1.

6.      What do the numbers in the new table 1 present? Is it average/median?

7.      The sentence “Others reported that papillary thyroid cancer with a thyroiditis background demonstrated much higher PD-L1 expression compared to papillary thyroid cancer with a normal background” needs to be referenced (line 323)

8.      The sentence “Several studies have demonstrated that during the dedifferentiation process in most thyroid cancer the expression of PD-L1 increases and NIS functional expression decreases.” needs to be referenced (line 343)

9.      Many lines in the manuscript have been repeated. Please, go carefully through all the manuscript and delete all the sentences that are repeated (saying the same thing).

For example:

in Line 335 - We also did not investigated PD-L1, or NIS expression in corresponding lymph node metastases, but others reported PD-L1 expression in metastatic papillary thyroid cancer tissues were similar to their corresponding primary tumor in the thyroid. [12].

In Line 380 - However, others have reported that PD-L1 expression in metastatic papillary thyroid cancer tissues is similar to that in their corresponding primary thyroid tumor [13].

Comments on the Quality of English Language

Some lines need to be edited

Author Response

Dear Reviewer,

thank you very much for the precious time you spent correcting our manuscropt.

We present the answeres to the comments.

1. The IHC staining of both markers is not presented in a uniform way, as requested.

The pathologist expert answered  this question as the following: in the Methods section we described the IHC scoring according  the pathologists experience.

  1. The authors said that the sentence “However, based on our study, the only conclusion that can be drawn is that there is not a correlation between the percentage of either NIS or PD-L1 expressing tumor cells in the primary tumor of lymph node metastatic PTC”, has been simplified, as requested. But they did not do it. There is still the line ‘However, based on our study, the only conclusion that can be drawn is that there is not a correlation between the percentage of either NIS or PD-L1 expressing tumor cells in the primary tumor of lymph node metastatic PTC’ in the abstract. In addition, this sentence is not clear, and it is insufficient. Please, delete it.

Authors deleted the requested sentences.

3. The authors did not reference the lines (between 47 and 63 lines in v1, and 62-73 in v2), as requested (although they said that they did it)

Authors cited the above mentioned lines.

4. Line 194-195 ‘Notably, there was no apparent correlation between PD-L1 and NIS expression (p=0.3199)’ should be transferred into the results section.

Transferred to the results section

5. There are two Table 1.

Authors deleted the original Table 1 and created a new one as requested by the other reviewer.

6. What do the numbers in the new table 1 present? Is it average/median

Corrected the Table 1. and inserted the mean(min-max)

7. The sentence “Others reported that papillary thyroid cancer with a thyroiditis background demonstrated much higher PD-L1 expression compared to papillary thyroid cancer with a normal background” needs to be referenced (line 323)

Referenced as requested.

8. The sentence “Several studies have demonstrated that during the dedifferentiation process in most thyroid cancer the expression of PD-L1 increases and NIS functional expression decreases.” needs to be referenced (line 343)

Referenced as requested.

9. Many lines in the manuscript have been repeated. Please, go carefully through all the manuscript and delete all the sentences that are repeated (saying the same thing).

The below mentioned lines were deleted.

For example:

in Line 335 - We also did not investigated PD-L1, or NIS expression in corresponding lymph node metastases, but others reported PD-L1 expression in metastatic papillary thyroid cancer tissues were similar to their corresponding primary tumor in the thyroid. [12].

In Line 380 - However, others have reported that PD-L1 expression in metastatic papillary thyroid cancer tissues is similar to that in their corresponding primary thyroid tumor [13].

Reviewer 3 Report

Comments and Suggestions for Authors

The manuscript has been improved, however, the Authors only partially addressed my comments/suggestions.

[1] I would suggest adding a scale bar to each image in Figure 1.

[2] The Authors still have not corrected the numbering of the Figures. They refer to Figures 3 (line 235) and 4 (line 236) in the main body of the manuscript, although no such Figures are present in the manuscript. Therefore, I strongly suggest that the Authors read Results section carefully and enter the correct Figure numbers. Moreover, it is also still not clear to me why the graphs showing the NIS expression data described in lines 236-240 ("Furthermore we found no correlation between NIS and the number of metastatic lymph nodes (p=0.6786), […] vascular invasion either (p=0.5945).” are not shown. I would be grateful for clarification. The font size in the Fig. 2 is too small - I would suggest using a larger font. 

[3] The Authors did not address my request, to mention other proteins (e.g., TPO) that are also important in the effective accumulation of radioactive iodine.

[4] Table 1: The Authors did not address my request to fill in the missing cells in Table 1 (the results for patients No. 54 and 70 are missing). If indeed these data are missing, please write ND, which stands for “not-determined” in each table cell.

[5] Table 1 legend: It seems to me that starting the legend with the following sentence would have made the message a little clearer: "Analysis of programmed death ligand 1 (PD-L1) and sodium-iodide symporter (NIS) expression in papillary thyroid cancer (PTC) sections detected by immunohistochemistry (IHC). ....". I would suggest slightly expanding the sentence "Range: 1=0%, 2=1-50%, 3= 51-100%" so that any Reader understands the meaning of the percentages.

[6] I would suggest referring to NIS as “sodium-iodide symporter” rather than “natrium-iodide symporter”.

[7] Lines 231-232: Please rephrase the following sentence “We examined PD-L1 (TPS) no correlation with the number of metastatic lymph nodes…”.

Examples of typos

[1] Lines 172, 177 and 178: Please change “is” to “was”.

[2] Line 174: Change “95 Celsius” to “95 degrees Celsius”.

Author Response

Dear Reviewer,

thank you very much for the valuable comments. We present our answers.

[1] I would suggest adding a scale bar to each image in Figure 1.

Scale bar is aadded to the Figure 1. to the left upper corners.

[2] The Authors still have not corrected the numbering of the Figures. They refer to Figures 3 (line 235) and 4 (line 236) in the main body of the manuscript, although no such Figures are present in the manuscript. Therefore, I strongly suggest that the Authors read Results section carefully and enter the correct Figure numbers. Moreover, it is also still not clear to me why the graphs showing the NIS expression data described in lines 236-240 ("Furthermore we found no correlation between NIS and the number of metastatic lymph nodes (p=0.6786), […] vascular invasion either (p=0.5945).” are not shown. I would be grateful for clarification. The font size in the Fig. 2 is too small - I would suggest using a larger font. 

Authors edited the Figures 2,3,4 to one Figure as Figure 2.

Authors clarified the figure legends.

[3] The Authors did not address my request, to mention other proteins (e.g., TPO) that are also important in the effective accumulation of radioactive iodine.

Authors mentioned other proteins.

[4] Table 1: The Authors did not address my request to fill in the missing cells in Table 1 (the results for patients No. 54 and 70 are missing). If indeed these data are missing, please write ND, which stands for “not-determined” in each table cell.

Authors edited a new Table 1.

[5] Table 1 legend: It seems to me that starting the legend with the following sentence would have made the message a little clearer: "Analysis of programmed death ligand 1 (PD-L1) and sodium-iodide symporter (NIS) expression in papillary thyroid cancer (PTC) sections detected by immunohistochemistry (IHC). ....". I would suggest slightly expanding the sentence "Range: 1=0%, 2=1-50%, 3= 51-100%" so that any Reader understands the meaning of the percentages.

The mentioned sentence is corrected as requested.

[6] I would suggest referring to NIS as “sodium-iodide symporter” rather than “natrium-iodide symporter”.

Corrected as sodium/iodide symporter 

[7] Lines 231-232: Please rephrase the following sentence “We examined PD-L1 (TPS) no correlation with the number of metastatic lymph nodes…”.

The sentence was rephrased.

Examples of typos

[1] Lines 172, 177 and 178: Please change “is” to “was”.

Corrected 

[2] Line 174: Change “95 Celsius” to “95 degrees Celsius”.

Corrected 

Round 3

Reviewer 2 Report

Comments and Suggestions for Authors

Dear authors,

Your paper "No Correlation between PD-L1 and NIS Expression in Lymph Node Metastatic Papillary Thyroid Carcinoma" has been sufficiently improved. However, there are still some remarks. Some are new, but some of them are still the previous ones that are not introduced in the most recent revision, despite your claim that you did:

1.        Figure 2 does not have a title. Please add it. Furthermore, it shows the associations of distribution only of PD-L1 (TPS) with the occurrence of invasion (capsular, vascular, lymphatic). In other words, there are no box plots of NIS, despite being named under the figure (in its legend). Therefore, boxplots of NIS should be added.

2.        The columns in Table 1 should be named.

3.        The correlation coefficients should be included to Table 2. Also, please clarify the acronym "Nr." in the table.

4.        There is a typographical mistake in line 266 (PTC). Please correct.

5.        Again, the authors did not reference the requested lines (between 55 and 64), although they said that they did it.

6.        Again, the manuscript contains repeated lines. Please go carefully through the whole manuscript again and delete all the sentences that are repeated (saying the same thing).

Author Response

Dear Reviewer,

Thank you again for your precious comments, we really appreciate them.

We followed your comments and instructions

1.Figure 2 does not have a title. Please add it. Furthermore, it shows the associations of distribution only of PD-L1 (TPS) with the occurrence of invasion (capsular, vascular, lymphatic). In other words, there are no box plots of NIS, despite being named under the figure (in its legend). Therefore, boxplots of NIS should be added.

We added  the title to Figure 2. NIS had no significant correlations, that is why we did not edited a table , only from TPS. In the results session we introduce the results regarding NIS.

  1. The columns in Table 1 should be named. We named it 
  2. The correlation coefficients should be included to Table 2. Also, please clarify the acronym "Nr." in the table. R is added in  the text in the result section in lines 206 and 207.
  3. There is a typographical mistake in line 266 (PTC). Please correct. papillary thyroid cancer -corrected
  4. Again, the authors did not reference the requested lines (between 55 and 64), although they said that they did it. We cited it as 2
  5. Again, the manuscript contains repeated lines. Please go carefully through the whole manuscript again and delete all the sentences that are repeated (saying the same thing). Deleted the lines which were repeated such as 288-295, 300-301,310-311, 315-316, 323-328.
